# Cellulose Fibre Degradation in Cellulose/Steel Hybrid Geotextiles under Outdoor Weathering Conditions

**DOI:** 10.3390/polym14194179

**Published:** 2022-10-05

**Authors:** Avinash Pradip Manian, Barbara Paul, Helene Lanter, Thomas Bechtold, Tung Pham

**Affiliations:** 1Research Institute of Textile Chemistry and Textile Physics, University of Innsbruck, Hoechsterstrasse 73, 6850 Dornbirn, Austria; 2Geobrugg AG, CH-8590 Romanshorn, Switzerland

**Keywords:** geotextiles, cellulose, biodegradation, moisture sorption, cellulase

## Abstract

Risks from rockfall and land sliding can be controlled by high-tensile steel nets and meshes which stabilise critical areas. In many cases, a recultivation of the land is also desired. However, high-tensile steel meshes alone are not always sufficient, depending on the location and the inclination of the stabilised slope, to achieve rapid greening. Cellulose fibres exhibit high water binding capacity which supports plant growth. In this work, a hybrid structure consisting of a nonwoven cellulose fibre web and a steel mesh was produced and tested under outdoor conditions over a period of 61 weeks. The cellulose fibres are intended to support plant growth and soil fixation, and thus the biodegradation of the structure is highly relevant, as these fibres will become part of the soil and must be biodegradable. The biodegradation of the cellulose fibres over the period of outdoor testing was monitored by microscopy and analytical methods. The enzymatic degradation of the cellulose fibres led to a reduction in the average degree of polymerisation and also a reduction in the moisture content, as polymer chain hydrolysis occurs more rapidly in the amorphous regions of the fibres. FTIR analysis and determination of carboxylic group content did not indicate substantial changes in the remaining parts of the cellulose fibre. Plant growth covered geotextiles almost completely during the period of testing, which demonstrated their good compatibility with the greening process. Over the total period of 61 weeks, the residual parts of the biodegradable cellulose web merged with the soil beneath and growing plants. This indicates the potential of such hybrid concepts to contribute a positive effect in greening barren and stony land, in addition to the stabilising function of the steel net.

## 1. Introduction

Serious concerns about micro-plastics in the environment have added credence to the constantly growing need to substitute petrol-based polymers with nature-based materials. Because of its abundant availability in nature, cellulose has been studied for many applications and its biodegradability is claimed as a positive argument in numerous publications [1].

Polymer degradation can be initiated by a chemical or physical attack to cellulose-based material and leads to significant losses in physical and mechanical properties. The chemical degradation of cellulose has been studied extensively to understand ageing of paper, where hydrolytic and oxidative reactions represent the main pathways in degradation [2]. The end-wise degradation (peeling) of cellulose also has been studied intensively under the conditions of regenerated cellulose fibre production [3,4,5]. Biodegradation occurs under the action of fungi, bacteria and algae via extracellular and intracellular enzymes and leads to complete mineralisation of the polymer [6]. In the case of the aerobic biodegradation of cellulose, the products are then CO_2_, water and biomass [7]. Important groups of extracellular enzymes are endo-β-1,4-glucanases and cellobiohydrolases (exo-β-1,4-glucanases), which lead to cleavage of the cellulose chain, preferably in amorphous regions, and release cellobiose [8,9]. The disaccharide is then hydrolysed into glucose by the action of β-1,4-glucosidase [8,10]. Additionally, in anaerobic cellulose biodegradation, the first step is the enzymatic hydrolysis to glucose and oligosaccharides, which are then digested by the anaerobic microbial community [11].

The biodegradation of cellulose-based textile products under simulated conditions has been demonstrated in the literature [12]. Lignin components were shown to have an inhibitory effect on the cellulolytic activity of cellulases, which is attributed to protein precipitation effects rather than to inhibitory effects on the cellulose-enzyme complex [13]. Additionally, the presence of finishing agents e.g., cationic softeners, can lead to retarded biodegradation of treated cellulose fibre-based textiles [14].

The biodegradability of a polymer depends on the physical and chemical properties of the material, as well as on the conditions under which biodegradation is considered. Thus, the conditions of biodegradation vary between composting, soil biodegradation, marine biodegradation, sewage sludge biodegradation, anaerobic biodegradation, landfill biodegradation and others [15]. As an example, lower biodegradation of cellulose fibres was observed under seawater conditions, in particular at 7 °C water temperature, when compared to conditions present during soil biodegradation [16].

The presence of cellulosic fibres in polyhydroxyalkanoates (PHA) leads to improved degradation behaviour under various conditions of degradation, thus indicating the positive influence of a readily degradable fibre on the biodegradation of the PHA matrix [15].

In soil burial tests, the degradation of cellulose fibres occurs via disintegration of the fibre structure and mineralisation of the fragments. In such wet soil, cellulose-based test fabrics were broken down completely within 4 weeks of biodegradation [17].

In forest soil, the degradation of cellulose is dependent on the type of soil, as well as on the climatic conditions, as a lower cellulose degradation rate was observed during dry seasons [18]. Additionally, the type of cellulose fibre determines the observed rate of fibre biodegradation, e.g., under warm, moist, aerobic soil conditions, more rapid degradation of viscose fibres was observed when compared to cotton and lyocell fibres [19]. Compared to the rapid biodegradation of the cellulose fibres in soil tests, no significant biodegradation was observed in the same tests for synthetic fibres, e.g., produced from polylactic acid (PLA), polyethylene terephthalate (PET) or polyacrylonitrile (PAN) [20].

Numerous standards to assess the biodegradability of a substance have been recommended; however, the specific conditions of the application of a product also determine the final observed biodegradation. Well-defined laboratory experiments and standardised biodegradation experiments permit access to reproducible results; however; the analysis of the specific behaviour of a polymer during biodegradation and the complex interaction with the environment requires the set-up of adequate field tests [21].

Depending on the test protocols used, full mineralisation of cellulose fibres into CO_2_ and H_2_O can reach values of 60–70%. This can value be understood as full mineralisation, as part of the fibre mass is also transformed with an increase in biomass [14].

The use of cellulose-based fibrous material as strengthening fibres in geopolymers has been extensively studied by researchers with a focus on strength, toughness of fibres against matrix cracking, degradation in the presence of chlorides, degradation by temperature and repeated wetting–drying cycles [22]. 

Non-woven structures, e.g., from polypropylene, are frequently used as the ground layer in the construction of roads to prevent road stones from sinking into the ground. Coarse jute-based fabrics are mounted as biodegradable geotextiles to stabilise steeper sites and to support their revegetation when only limited mechanical stress is expected to occur. Polypropylene, polyethylene and polyethyleneterephthalate-based products have been recognised as sources for microplastics, while the biobased cellulose fibres viscose, modal and lyocell exhibit biodegradability. Differences in the polymer packing and order between viscose, modal and lyocell, as well as cotton modulate the rate of biodegradation, however, do not affect the biodegradability of the cellulose fibres (Röder et al., 2019 [23]).

High-tensile steel nets and meshes are frequently used to secure landslide prone slopes above roads and railways, for example. Such structures provide mechanically excellent long-term stabilisation of critical sites. Depending on the inclination and orientation of said slope, revegetation occurs naturally.

However, when rapid vegetation of an area is at risk because of the steepness of the slope, the orientation is unfavourable because it dries out the soil, or when there is a risk of soil erosion through water percolation, a combination of the steel nets with geotextiles is necessary to achieve revegetation. There are two types of geotextiles that can be used depending on the site. When revegetation might take several years to settle definitively, a long-term geotextile is requested which does almost not degrade in the first years. On the other hand, many sites are easily revegetated within a season’s cycle if the right support of initial plant growth is given, and the lifetime of the geotextile layer does not need to exceed this period. This can be achieved, for example, with water sorbing cellulose fibre webs.

While the steel net provides mechanical stability and safety over many years, the cellulose fibre web should contribute to the formation of humus and greening. Such geotextiles are frequently mounted in an ecologically sensitive environment; thus, knowledge about interaction with plant growth and biodegradability of the fibre web under conditions of simulated application are required to understand the behaviour of such fibres during biodegradation. In this application, the cellulose fibres form the first surface layer on the ground, thus a specific setup to monitor biodegradation under real conditions is required to assess the behaviour of the cellulose fibres in the environment.

In this particular application, the cellulose part of the hybrid geotextiles should stabilise the ground during the period of plant growth followed by a phase of biodegradation and integration into the soil. At present, no scientific results about the biodegradation of cellulose fibres when positioned at the surface of the ground have been reported in the literature. In this paper, a long-term study to monitor the biodegradation of cellulose in cellulose/steel hybrid geotextiles under field test conditions is presented. Larger dimension prototypes (1.3 × 1.3 m²) were prepared, mounted on the ground and monitored over a period of 427 days. The prototypes were prepared in cooperation with a viscose fibre producer, thus viscose fibres were used for this study. Samples of the cellulose fibres were taken in regular intervals and characterised by the analytical techniques: Fourier transform infrared spectroscopy (FTIR), moisture sorption, viscometry, laser scanning microscopy and carboxyl group content to monitor fibre degradation during the period of observation. The set of elaborated data contributes significant information to the complex behaviour of cellulose fibres in the environment and also describes the behaviour of the new cellulose/steel hybrids under real growth conditions.

## 2. Experimental

### 2.1. Materials and Chemicals

Throughout this work, 100% viscose fibres (CV) with a linear density of 1.7 dtex, kindly supplied from Kelheim Fibres GmbH (Kelheim, Germany), were used as received. Norafin Industries GmbH (Mildenau, Germany) manufactured the nonwoven material made from the viscose fibres. For the preparation of the geo textiles, the steel mesh DELTAX^®^ G80/2 from Geobrugg AG (Romanshorn, Switzerland) was used and two starch products were used as fibre binding agents: Maltodextrin starch (S1) and corn starch (S2) (Maizena, Unilever Austria GmbH, Wien, Austria).

Methylene blue (MB), borate, HCl, FeCl_3_∙6H_2_O, tartaric acid and NaOH were analytical grade chemicals.

### 2.2. Production of Geo Textile Prototypes/Wet-Laying Process

The prototypes were manufactured as a combination of steel mesh, viscose fibre and viscose hydroentangled nonwoven in a size of 130 × 130 cm². The starch solution was added to strengthen the bonding between open CV fibres, the CV nonwoven and supported the anchoring of the cellulose web to the steel fabric. 

In a first step, the steel mats were cut to size and, depending on the type of construction (Table 1), a selected nonwoven fabric of the same size was placed underneath. Un-crimped fibres with an approximate length of 100 mm were opened by hand and placed loosely on the steel mesh. 

A wet-laying system was built (Figure 1) that allows water to continuously drain through the floor and filter out the fibres to form a matted web, which is then dried to fix the fabric. The starch was dissolved in hot water and evenly distributed on the layer. The prototypes were removed from the frame after a drying time of about 48 h.

### 2.3. Biodegradation Tests

For the degradation tests and the investigation of the biological degradation of the cellulose fibre structure, the geotextiles were mounted outdoors on a small slope with full sun exposure near the location of the producer Geobrugg in Romanshorn in Switzerland (Elevation: 406 m (1332 ft) in summer 2019. Accurate location of the experimental site was defined by GPS coordinates (47°33’24.4” N, 9°21’59.9” E). Fibre samples were taken at regular intervals and brought to the laboratory for analysis. 

### 2.4. Determination of Moisture Content and Carboxyl Group Content

The collected cellulose samples were conditioned at 21 ± 2 °C at 65 ± 2% RH for at least 24 h. To determine the exact dry weight, the samples were weighed and dried at 105 °C overnight. They were then cooled in a desiccator with P_2_O_5_ and weighed in dry state to determine the moisture content from the weight loss. The values are given as the mean of a double determination [24]. 

The methylene blue sorption measurements were carried out with 0.5 g of dry sample [25,26]. First, 0.30 g methylene blue was dissolved in 750 mL deionised water. Then, the pH was adjusted to 8.5 by the addition of NaOH and the solution was filled to 1 L. Two samples were analysed for each specimen. The dried sample was chopped and mixed with 10 mL of 0.30 g L^−1^ MB solution and 25 mL borate buffer pH 8.5 and then made up to 50 mL with deionised water. After 1 h of stirring at room temperature, the suspensions were filtered through a glass filter. Then, 10 mL of the filtrate was diluted in 10 mL 0.1 M HCl and made up to 50 mL with deionised water. The diluted solution was analysed by UV–Vis spectrophotometry at a wavelength of 664.5 nm. (Spectrophotometer HITACHI U-2000, Inula, Vienna, Austria, 10 mm cuvette) Borate buffer served as reference. The absorbance of the solution was used to determine the total content of acid sites in the geotextiles. The carboxyl group content was then calculated as mmol kg^−1^ cellulose material. To ensure the repeatability of the methylene blue results, the absorbance was additionally measured using 5 mL of the filtered suspension. The results were given as mean value and the standard deviation of six independent experiments.

### 2.5. Determination of Viscosity-Average Degree of Polymerisation

A viscometric determination of the average degree of polymerization (average DP_v_) was performed using an Ubbelohde capillary viscometer (Type No 501, Capillary 12/Ic) (SI Analytics GmbH, Mainz, Germany) and fresh ferric sodium tartrate (FeTNa) solvent (0.3 M FeCl_3_∙6H_2_O, 0.98 M tartaric acid, and 5.36 M NaOH). Cellulose samples (20 mg) were dissolved in 50 mL FeTNa solution. Argon gas was purged into the bottle to remove air and prevent oxidation during dissolution. The bottle was closed and shaken for more than 16 h to make sure that cellulose was completely dissolved. The viscosity η of a cellulose solution was measured from the efflux time at 20 ± 1 °C through the viscometer and the intrinsic viscosity was calculated by equation (DIN 54270-3). The results are given as mean and standard deviation of a double determination.
(1)$$\eta =\frac{\left(\eta_{\mathrm{rel}}-1\right)/c}{1+k\cdot \left(\eta_{\mathrm{rel}}-1\right)}$$

η = intrinsic viscosity (mL g^−1^)

c = concentration of viscose sample in the solution (g mL^−1^)

k = 0.339 

$\eta_{\mathrm{rel}}=\frac{t}{t_{0}}$ with t (in seconds) and t0 being the efflux time of the solution and the solvent respectively.

The viscosity average DPv is calculated from the intrinsic viscosity η using Equation (2) [27].
(2)$$\eta =4.85\times {\mathrm{DP}}_{v}^{0.61}$$

### 2.6. Confocal 3D Laser-Scanning Microscopy and FTIR Spectroscopy

The nonwoven fabric samples were examined with a confocal 3D laser-scanning microscope (Keyence VK-X150, Keyence, Neu-Isenburg, Germany) at 20×, 50× and 100× magnification, using the on board software.

The Fourier transform infrared spectra (FTIR) of the fibres were recorded in the spectral range 4000 cm^−1^ to 500 cm^−1^ using an attenuated total reflectance (ATR) unit equipped with a diamond crystal (Bruker Vector 22, Karlsruhe, Germany; resolution 2 cm^−1^, and 128 scans). 

## 3. Results and Discussion

### 3.1. Exposure to Weathering and Visual Inspection

Biodegradation of the cellulose fibres under real weathering conditions depends on the actual climate conditions during the exposure. The prototypes were installed directly on the ground on a slope in Romanshorn (CH) (Elevation: 406 m, 1,332 ft, GPS coordinates 47°33’24.4” N, 9°21’59.9” E). Bacterial growth and plant growth in the structure depend on the actually prevalent temperature, precipitation and season, respectively, thus the climate data given in Figure 2 provide an overview about most important conditions during the 61 weeks degradation test in the moderate climate of the region. The simulated raw data were provided by meteoblue (meteoblue.com, accessed on 2 August 2022). Red triangles in the diagram show when the samples were taken for laboratory analysis.

Table 2 shows the detailed sampling time and the overall duration of the exposure to outdoor weathering conditions. 

Over the period of 61 weeks, an accumulated total amount of precipitation of 900 mm water was calculated. Precipitation occurred in quite regular intervals without a long dry period. The temperature mainly ranged from 5 °C to 20 °C, with a minimal temperature of −4 °C in winter and a maximum temperature of 30 °C during summer.

The biological degradation of the cellulose web was also monitored by visual and microscopic inspection and with chemical analytical methods.

The biological decomposition of the materials can be easily assessed visually. Photographs of the prototypes were taken at the same time when material samples were collected for laboratory analysis. The appearance of the geotextiles during the weathering is shown in Figure 3 and Figure 4, using photographs of prototype 1. The integration of the geotextiles into nature began quite quickly: after 11 weeks, about half of the geotextile was already covered with vegetation and firmly anchored to the soil. The exposure to sun and rain during the first period of the installation (August–November), along with the open structures in the geotextile, ensured the rapid development of the vegetation and overgrowth with plant material. After 35 weeks, the prototypes were almost completely overgrown with vegetation; however, the cellulose web can still be recognised in a close-up view (Figure 3f).

The process of degradation was also indicated by the dark discolouration of the sample pieces in Figure 4. Rapid progress of plant growth and integration of the chosen geotextile structure also accelerates the access of microorganisms. The results show that the degradation is also co-initiated by soil microorganisms, as with longer degradation time, the brown discolouration intensifies [20]. The weather trials started in the summer with higher temperatures, followed by the wet autumn season; both factors also accelerate the growth of microorganisms and, thus, fibre degradation. 

The biological attack and degradation also led to changes in the flexibility of the material. The nonwoven structures became more brittle and thinner over the weeks and completely fragmented at the end of the trials (Figure 5). As a result, sampling of pure cellulose fibre containing specimen became impossible and the presence of soil and plant residues made determining the degree of polymerisation impossible. A photodocumentation of the degradation of the 6 prototypes is given in the Appendix A.

These observations already indicate the rapid decomposition of the chemical and morphological structure of the fabric samples within the test period.

Figure 6 shows the photomicrographs of samples after different times of weathering. It can be observed that microorganisms settled on the surface of the fibres. After more than a year of weathering tests, the geotextiles were noticeably degraded, and the integrity and structure of the fabric almost collapsed. Remarkably, the remaining material still exhibited the shape of a fibrous structure and very little fibrillation or fibre fragments was detected in the photomicrographs. Most probably, the fibres biodegraded from their more accessible fibre ends, where enzymatic hydrolysis then leads to the release of cellobiose and glucose oligomers. This is in agreement with the proposed mechanism for enzymatic cellulose degradation, where amorphous zones of the cellulose fibre are hydrolysed more rapidly. Free chain ends are formed by the endo-glucanases and are then hydrolysed to cellobiose by exo-glucanases [6].

### 3.2. Monitoring of Fibre Degradation

While visual inspection and microscopic analysis of the prototypes during the period of outdoor testing provide information about physical damage, structure loss and plant growth, analytical methods have to be applied to monitor cellulose biodegradation on a molecular level. An important measure indicating cellulose degradation is obtained from measurements of the dynamic viscosity of the cellulose after dissolution in ferric sodium tartrate (FeTNa) solvent.

In case biodegradation of the cellulose polymer occurs in the fibre structure, a reduction in the dynamic viscosity of the solution should be detected when compared to pristine viscose fibres. In Figure 7, the change in the DP_v_ values as a function of weathering time are given. DP_v_ changes from about 400 for after 60 weeks to 100 for the cellulose fibres after a degradation period of 22 weeks. After longer weathering time (32 weeks or more), the fibres could no longer be dissolved completely by ferric sodium tartrate (FeTNa) solvent, which also could contribute to an apparently lower DP_v_ value. The DP_v_ value of the pristine viscose fibres at the beginning ranges at 400 ± 53 glucose units per cellulose molecule. Up to 11 weeks of weathering, the reduction in DP_v_ value is not significant. After 22 weeks of outdoor weathering, a reduction to DP_v_ of 200 is indicated by the measurement of the dynamic viscosity. The remarkably low reduction in DP_v_ can be explained with the mechanism of biodegradation. Where small fragments of cellobiose and glucose oligomers are removed from the fibre surface as products from the enzymatic hydrolysis, the DP_v_ of the cellulose in the remaining fibre bulk will not be affected substantially. 

The results for DP_v_ measured at the samples taken after 11 weeks of outdoor degradation indicate a slight increase in DP_v_, which, however, must be interpreted under consideration of the relatively high standard deviation. To bring the result to a statistically significant level, a very high number of additional repetitions would be required.

### 3.3. Determination of Moisture Content and Carboxyl Group Content

The collected samples were characterised by determining their moisture content and by analysing the carboxylic group content. 

The initial moisture content of the viscose fibre was determined with 12 %wt, which is the expected value for regenerated cellulose fibres produced by the xanthogenate route. Following a model of structural weakening and opening of the fibre structure, an increase in moisture sorption during biodegradation would be expected initially. Remarkably, the moisture content decreased during the phase of biodegradation. The decrease from 12.5 ± 0.5 %wt at week 6 to 11.1 ± 0.5 %wt after week 32 is statistically significant at a two-sided confidence level of 99.9%. This finding can be understood on the basis of the enzymatic processes, which lead to more rapid degradation of the amorphous parts of the fibre. As the amorphous parts of a cellulose fibre are responsible for moisture sorption biodegradation processes, which reduce the share of the amorphous regions preferentially, it will lead to a reduced sorption capacity for water of the remaining fibre structure. The overall change in moisture sorption of the cellulose fibres as a function of time is shown in Figure 8. A few samples exhibit a rather high standard deviation, which is indicated by the comparable high error bars. This can be explained with the presence of other impurities, e.g., soil, plant material, which could not be removed completely from the specimen.

During viscose fibre manufacturing, the reducing end of the cellulose chain is oxidised and, instead of the aldehyde group, carboxylic groups are formed. Representative values for the carboxylic group content of viscose fibres are near 20 mmol kg^−1^ fibre material [28]. The carboxylic group content in viscose fibres used for these experiments was determined with 17.3 ± 0.1 mmol kg^−1^. 

During the biodegradation of the cellulose fibres, weathering hydrolysis of the glycosidic bonds in the cellulose chains and oxidation of the formed aldehyde groups could occur. As a consequence, an increase in carboxylic group content should be observed. The carboxylic group content in the collected samples as a function of weathering time is given in Figure 9.

The analysed content of the carboxylic groups remained constant over the period of outdoor weathering, which indicates that light-induced cellulose chain breakage followed by photo-oxidation of the end groups formed to carboxylic groups is a minor pathway, compared to the enzymatic degradation processes. The higher scattering of the results with increasing time outdoors can be explained with the sensitivity of the methylene blue sorption method to other impurities present in the specimen. The presence of plant residues and microorganisms on the fibre specimen (Figure 6), as well as soil residues, can lead to higher variations in methylene blue sorption and, thus, increase the variation in experimental results.

FTIR-ATR analysis of the samples was performed to identify the appearance of new functional groups, e.g., carbonyl groups or carboxyl groups as the results of the weathering tests. Representative spectra of samples after different time in outdoor weathering are shown in Figure 10. A zoomed-in portion of the fingerprint region is shown in Figure 11.

The spectra shown in Figure 10 exhibit the typical absorbance pattern for cellulose. A detailed table of characteristic peak values is shown in Table 3.

The broad absorbance between 3500 cm^−1^ and 3000 cm^−1^ can be attributed to the stretching vibration of the O-H groups and the absorbance near 2900 cm^−1^ can be attributed to the C-H stretching vibration. C-H and CH_2_ bending vibrations are observed from 1500 cm^−1^ to 1100 cm^−1^, and the C-O stretching vibration leads to the strong absorbance at 1100 cm^−1^ [29]. Only minor differences are seen in the FTIR spectra, with longer exposure time in outdoor weathering. Smaller variations in the wavenumber region 2750 cm^−1^–3000 cm^−1^ (C-H stretching vibrations) may be due to the presence of higher amounts of microorganisms, as well as adhering soil residues.

In the FTIR of the sample collected after 35 weeks, a smaller signal is detected around 1750 cm^−1^, which most probably is due to the presence of other material (plant material, microbial growth) at the particular site where the FTIR spectrum was measured. At this wavenumber, ester groups, e.g., present in membrane lipids and cell wall pectines, are observed [32].

## 4. Conclusions

Hybrid structures made from cellulose fibre nonwovens and steel wire provide two complementary functions when applied as a combination of geotextiles and slope stabilisation solutions. The steel mesh contributes with the required physical strength to prevent rock fall and land sliding. The hydrophilic viscose fibres serve as a biodegradable water reservoir, which supports greening in a stabilised area. During the period of greening, biodegradation of the cellulose fibres may occur. 

A long-term outdoor study with prototypes of the geotextiles was executed with a total duration of 61 weeks of weathering to monitor the biodegradation of the cellulose fibres and plant growth. Samples of the cellulose nonwovens were collected regularly over the test period and the biodegradation was monitored by microscopy and instrumental analytical methods. The carboxylic group content of the residual material did not change during the period of weathering and the formation of additional functional groups, e.g., carbonyl groups, was not indicated by FTIR analysis. However, a significant decrease in the moisture content was observed with the duration of the outdoor weathering. These findings are in agreement with the theory of enzymatic degradation of cellulose fibres [6]. The biodegradation of cellulose occurs via the action of cellulases, which hydrolyse the cellulose chains and leads to the release of cellobiose and anhydroglucose oligomers. This reaction preferentially occurs at the amorphous domains of the cellulose fibres. The amorphous domains are responsible for the moisture sorption in the fibres; thus, degradation of these domains will also lead to reduced water sorption behaviour. A reduction in the average chain length of cellulose was also detected by determining the degree of polymerisation. During the phase of biodegradation, the nonwoven cellulose structure partially disintegrated and plant growth through the nonwoven structure occurred. 

The results demonstrate that the desired functionality of a short-term biodegradable flexible structure with a durable steel backbone, which supports soil fixation, contributes to water storage and enables plant growth successfully, could be achieved with the cellulose/steel hybrid structure.

Future research could address the preparation of regenerated cellulose fibres from cellulose containing wastes, e.g., from flax production as well as from recycled cellulose material, to reduce the ecological impact of fibre production [32,33]. Such a comparison should also include the different processes for fibre production, e.g., the lyocell process and the use of ionic liquids.

## Figures and Tables

**Figure 1 polymers-14-04179-f001:**
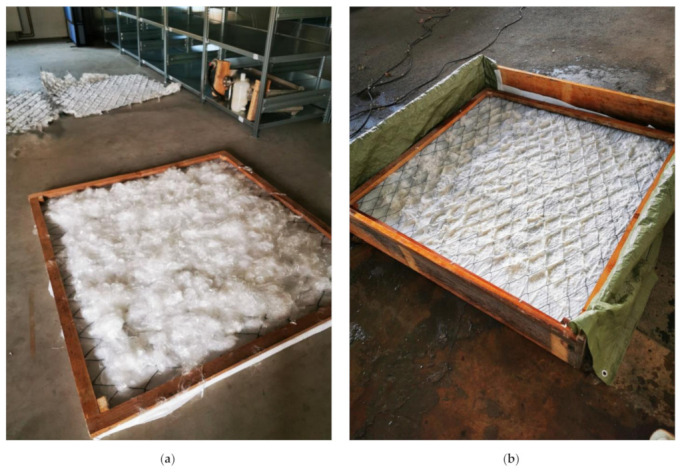
Preparation of a prototype, (**a**) Viscose fibres are distributed randomly on the steel mesh, underneath there is a viscose fleece, (**b**) prototype after wet-laying process.

**Figure 2 polymers-14-04179-f002:**
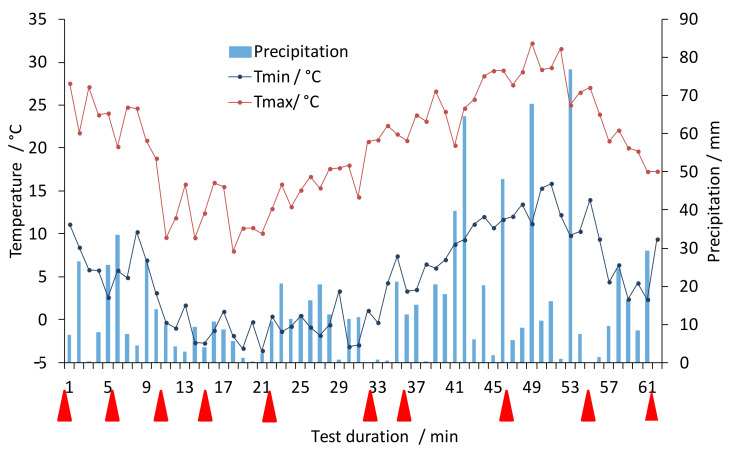
Weather data simulated for the site of the out-door prototype testing (Romanshorn, CH) during the period of outdoor degradation tests (data provided by meteoblue.com).

**Figure 3 polymers-14-04179-f003:**
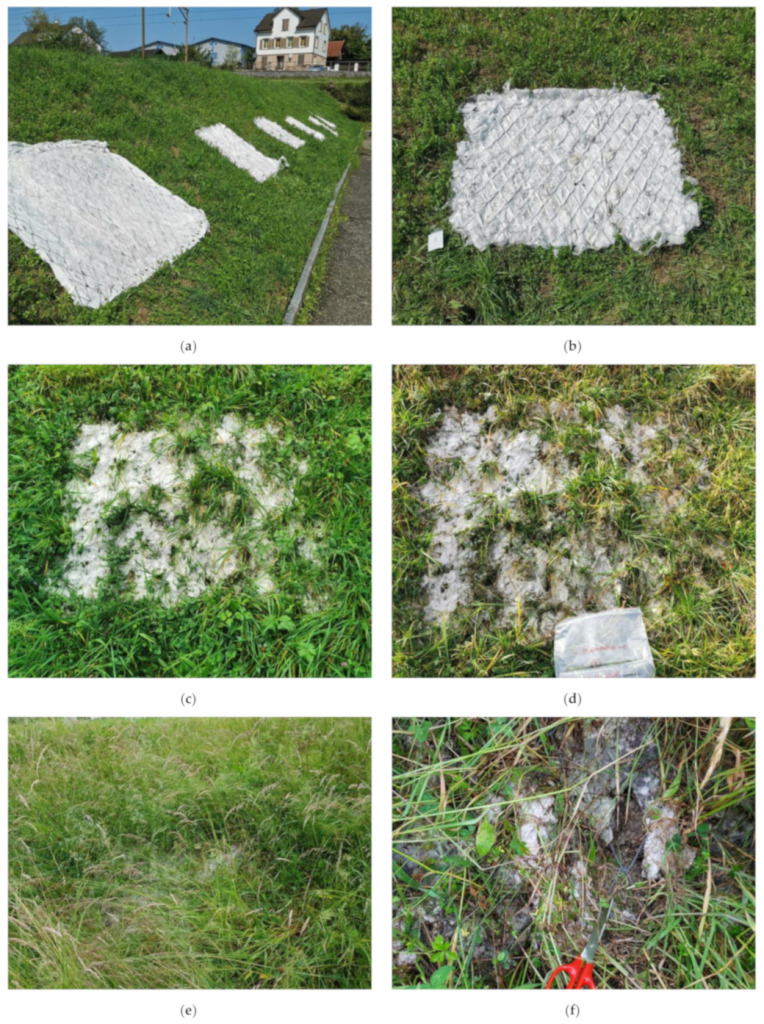
Photographs of the installed prototypes, (**a**) installed prototype 1–6, (**b**) day of installation, (**c**) after 11 weeks, (**d**) after 22 weeks, (**e**) after 35 weeks (**f**) close-up view after 35 weeks.

**Figure 4 polymers-14-04179-f004:**
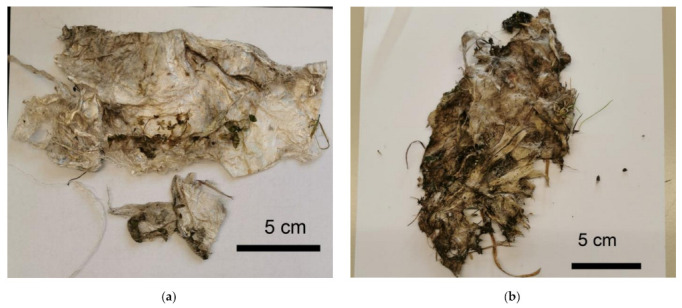
Photographs of a cellulose fibre sample from prototype 1, (**a**) 15 weeks, (**b**) 47 weeks of outdoor installation.

**Figure 5 polymers-14-04179-f005:**
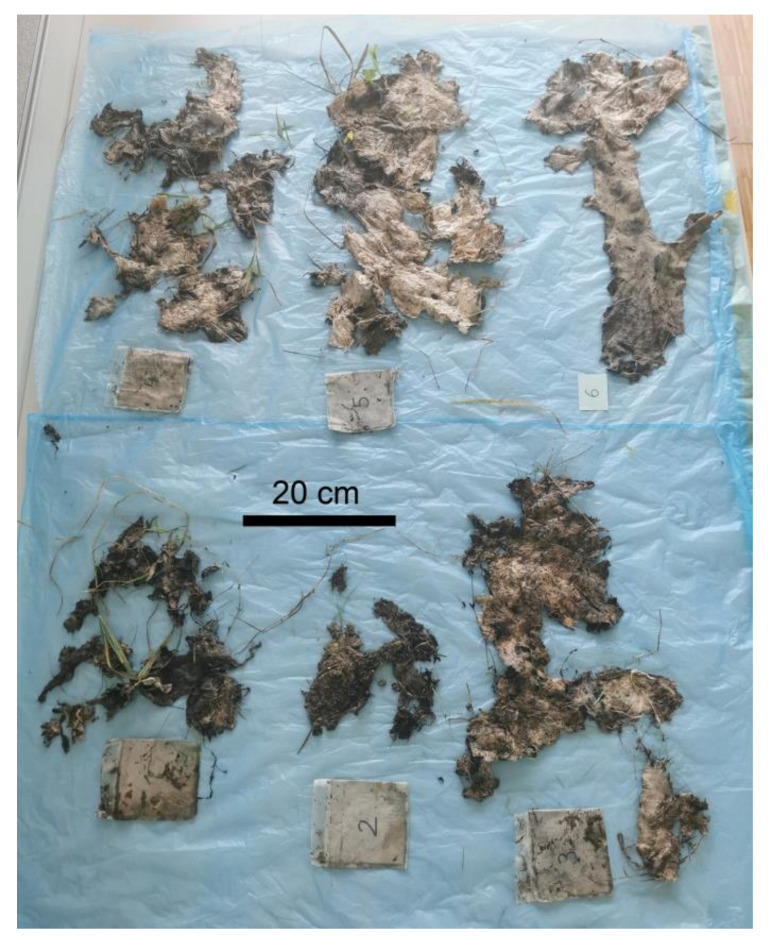
Photographs of samples from prototype 1–6 after 61 weeks of outdoor weathering.

**Figure 6 polymers-14-04179-f006:**
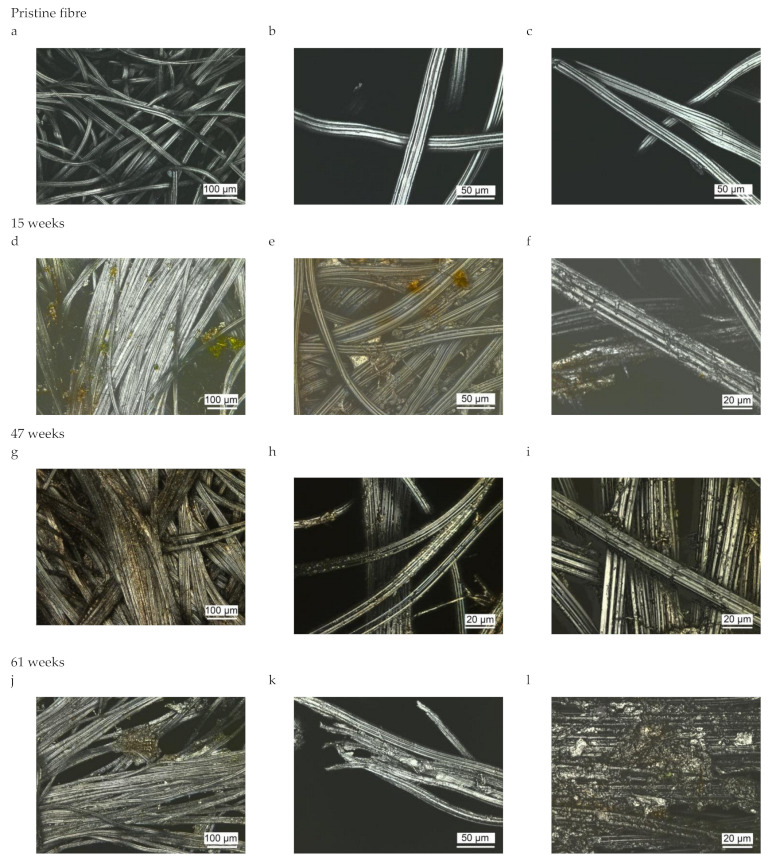
Photomicrographs of viscose fibres as function of time in outdoor weathering. (**a**) pristine CV fibres, (**b**,**c**) with use of higher magnification; (**d**) fibres after 15 weeks of outdoor weathering, (**e**,**f**) with use of higher magnification; (**g**) fibres after 47 weeks of outdoor weathering, (**h**,**i**) with use of higher magnification; (**j**) fibres after 61 weeks of outdoor weathering, (**k**,**l**) with use of higher magnification.

**Figure 7 polymers-14-04179-f007:**
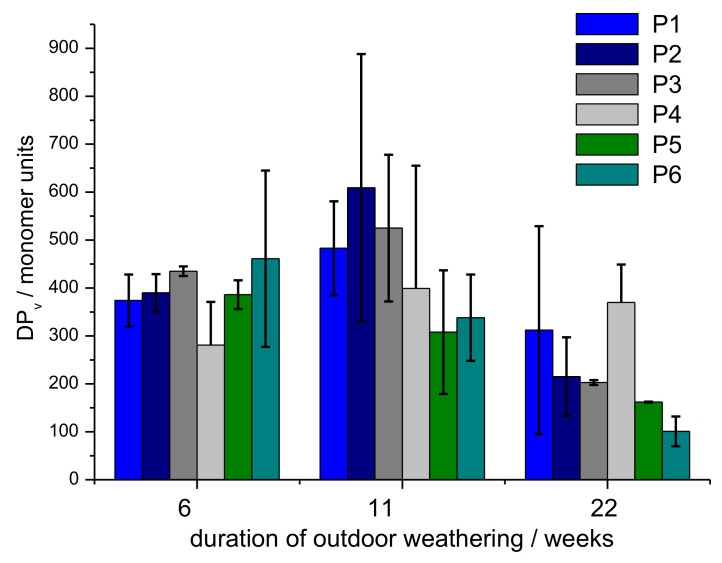
Degree of polymerization of the cellulose as function of time under outdoor conditions.

**Figure 8 polymers-14-04179-f008:**
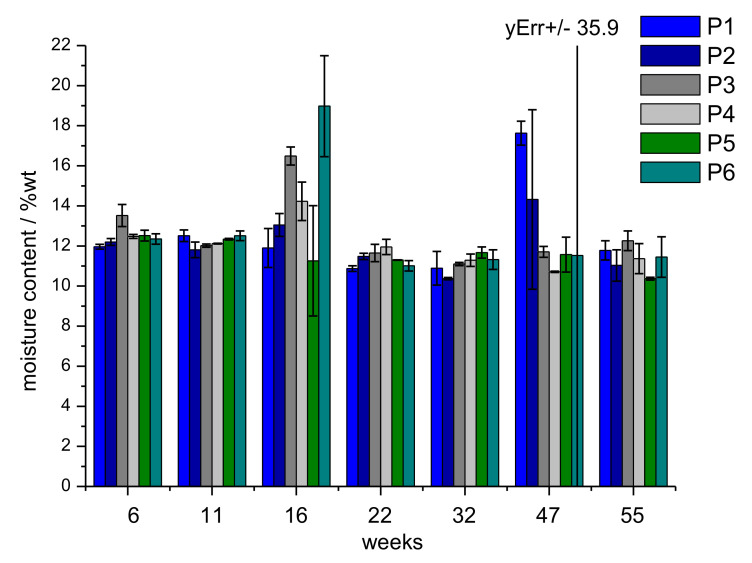
Moisture content of the viscose fibres of the prototypes during outdoor weathering as function of time (mean and standard deviation as error bar).

**Figure 9 polymers-14-04179-f009:**
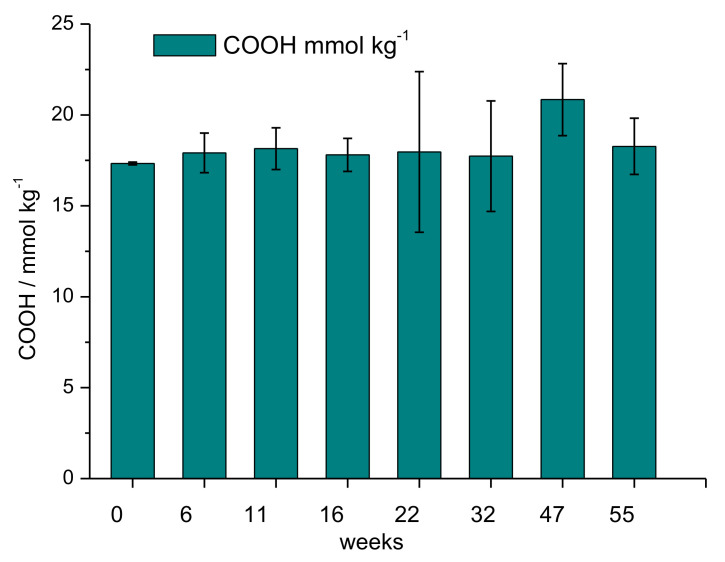
Carboxyl group content of weathered viscose samples as function of time.

**Figure 10 polymers-14-04179-f010:**
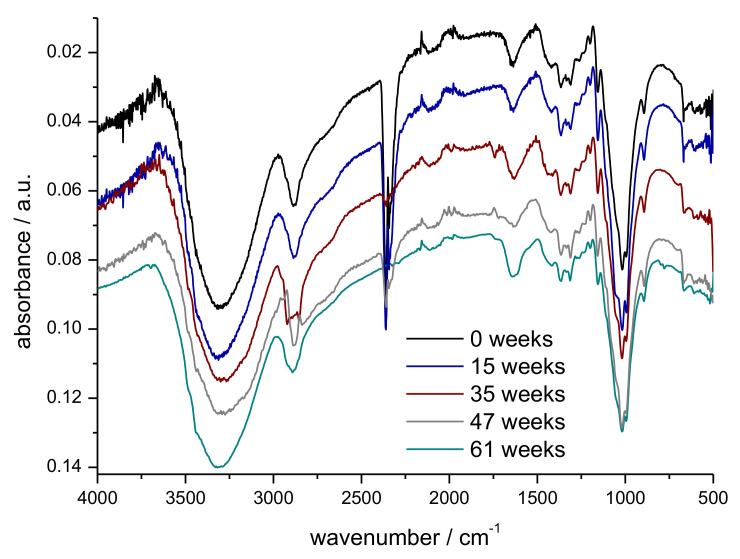
FTIR-ATR spectra of prototype 4 as function of exposure time in outdoor weathering (week 0, 15, 35, 47, 61).

**Figure 11 polymers-14-04179-f011:**
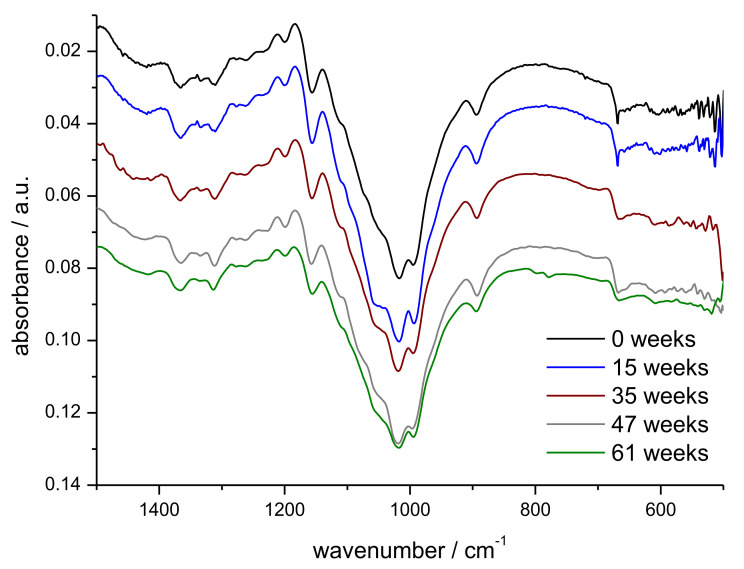
Zoomed-in section of the fingerprint region of theFTIR-ATR spectra of prototype 4 as function of exposure time in outdoor weathering (week 0, 15, 35, 47, 61).

**Table 1 polymers-14-04179-t001:** Compositions of various geotextile prototypes for the degradation tests (binder: S1 maltodextrin, S2 corn starch).

Prototype	Mass per Area Viscose Fibres g m^−2^	Mass per Area Nonwoven Fabric g m^−2^	Ratio of Binding Agent per Fibre, Binder g m^−2^
P 1	200	200	0.25 S1
P 2	200	200	0.25 S1
P 3	200	200	0.85 S2
P 4	200	200	0.25 S2
P 5	200	**-**	0.25 S2
P 6	130	200	0.25 S2

**Table 2 polymers-14-04179-t002:** Period of the degradation tests and sampling dates.

		Month	Weeks	Days
	Start of weather trials	August	0	1
1	sampling	October	6	42
2	sampling	November	11	77
3	sampling	December	15	108
4	sampling	January	22	157
5	sampling	April	32	228
6	sampling	May	35	248
7	sampling	July	47	335
8	sampling	September	55	365
9	End of weather trials	October	61	427

**Table 3 polymers-14-04179-t003:** Characteristic absorption peaks in the infrared spectrum of the viscose fibres and literature reference values for cellulose [29,30,31].

Wavenumber	Literature	Type of Vibration
cm^−1^	cm^−1^	
3000–3500	3000–3500	O-H hydrogen bonded stretching
2926–2886	2892	C-H stretching
1417–1420	1430	H-C-H and O-C-H in plane bending
1365–1366	1375	C-O-C, C-C-O, C-C-H deformation and stretching
1310–1313	1312	O-H bending
1156–1157	1157	C-O asym. valence
1018	1026	C-O-C pyranose ring skeletal
893	892	C-O-C valence
666–668	668	C-OH out of plane bending
1636, 1641	1635, 1638	Adsorbed water

## Data Availability

The data presented in this study are available on request from the corresponding author. The data are not publicly available due to confidentiality reasons.

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
