# Peer review of "Cellulose Fibre Degradation in Cellulose/Steel Hybrid Geotextiles under Outdoor Weathering Conditions"

_polymers, 2022, doi:10.3390/polym14194179_

Round 1

Reviewer 1 Report

The manuscript of Thomas Bechtold et al is devoted to the study of the properties of hybrid systems based on metal meshes and geotextiles in natural conditions. Emphasis is placed on the behavior of cellulosic textiles. After reviewing the theoretical part, the question immediately arises, what is the novelty of the work? The authors themselves note a number of papers (although not a large list) that describe the behavior of cellulose products in the environment. If I'm not mistaken, similar work was also carried out by Lenzing (T. Roder et al), I did not find references to these studies in the work and I recommend studying these works. Next, the authors should point out the differences in the behavior of geotextiles in the presence and absence of metal products.     Lines 114, 115. "In this paper a long term study to monitor the biodegradation of cellulose/steel hybrid geotextiles" - here it is better to just talk about cellulose. Table 2. Why was the experiment started in August? Do construction work usually start in spring (early summer)? Figure 6. Increase the font size on the scale bar. The FTIR data is presented in a very compressed form. For example, what happens to samples at 35 and 47 weeks (region 2750-3000 cm-1). Figure 10. Specify the dimension.   References. Link numbering missing.   In my opinion, the work is more suitable for the Materials journal (MDPI) than for Polymers. Also, in my opinion, the scientific novelty of the work is not obvious!

Author Response

Response to reviewers polymers-1910437

Cellulose fibre degradation in cellulose/steel hybrid geotextiles under outdoor weathering conditions

Avinash P. Manian, Barbara Paul, Helene Lanter, Thomas Bechtold and Tung Pham

Dear Editor,

First we want to thank the reviewers for the careful revision. We considered all recommended changes and hope the manuscript now is acceptable for publication. We have highlighted the revised elements in colour.

Please find below the list of changes.

Thomas Bechtold

Reviewer 1

After reviewing the theoretical part, the question immediately arises, what is the novelty of the work?

Action: We have added more information about the novelty of the study at the introduction section.

The authors themselves note a number of papers (although not a large list) that describe the behavior of cellulose products in the environment. If I'm not mistaken, similar work was also carried out by Lenzing (T. Roder et al), I did not find references to these studies in the work and I recommend studying these works.

Action: We added appropriate references of T. Roeder and also further reference (A. Potthast) with regard to the cellulose degradation

Next, the authors should point out the differences in the behavior of geotextiles in the presence and absence of metal products.

Action: We added discussion of representatives for different geotextiles

Lines 114, 115. "In this paper a long term study to monitor the biodegradation of cellulose/steel hybrid geotextiles" - here it is better to just talk about cellulose.

Action: We reformulated to emphasise the focus on cellulose.

Table 2. Why was the experiment started in August? Do construction work usually start in spring (early summer)?

Comment: The installation was timed in agreement with the company who is a specialised producer for such steel geotextiles (pls. see co-author)

Figure 6. Increase the font size on the scale bar.

Action: We increased font size on scale bars in Figure 6

The FTIR data is presented in a very compressed form. For example, what happens to samples at 35 and 47 weeks (region 2750-3000 cm-1).

Action: We added more detailed discussion

Figure 10. Specify the dimension.

Action: We added dimensions to Figure 10

  References. Link numbering missing. In my opinion, the work is more suitable for the Materials journal (MDPI) than for Polymers.

Comment: We are convinced that the material will be a valuable contribution to the special issue “Natural Degradation: Polymer Degradation under Different Conditions”

Also, in my opinion, the scientific novelty of the work is not obvious!

Action: We extended the description of novelty in the introduction part.

Reviewer 2 Report

This article clearly describes that hybrid structures made from cellulose fibre and steel may be helpful in various applications such as soil fixation, and water storage. The article is well written; however, some modifications are suggested as below:

1.     Improve the fig captions. In Fig 8,9. The x axis should be weeks. The std dev is too high in some cases and does not show an upper limit in Fig 8.

2.     In FTIR spectra, the author should include the units for both the x and y axis

3.     It would be helpful if the authors could include a reference table along with exact peak values with respect to the FTIR spectra in Fig. 8.

4.     It would be helpful if a zoomed-in portion of the fingerprint region can be represented.

5.     The FTIR spectra should be baseline subtracted and deconvoluted mandatorily for better representation and understanding. There are several peaks that are not explained in the text or discussion.

6.     Please mention the magnifications used for optical images.

7.     Is there any reference for 2.3 Biodegradation tests, and 2.4. Determination of moisture content and carboxyl group content? If so, please add it.

8.     In general, how many times each experiment was repeated? author should mention.

Author Response

Response to reviewers polymers-1910437

Cellulose fibre degradation in cellulose/steel hybrid geotextiles under outdoor weathering conditions

Avinash P. Manian, Barbara Paul, Helene Lanter, Thomas Bechtold and Tung Pham

Dear Editor,

First we want to thank the reviewers for the careful revision. We considered all recommended changes and hope the manuscript now is acceptable for publication. We have highlighted the revised elements in colour.

Please find below the list of changes.

Thomas Bechtold

Reviewer 2

Comments and Suggestions for Authors

This article clearly describes that hybrid structures made from cellulose fibre and steel may be helpful in various applications such as soil fixation, and water storage. The article is well written; however, some modifications are suggested as below:

  1. Improve the fig captions. In Fig 8,9. The x axis should be weeks. The std dev is too high in some cases and does not show an upper limit in Fig 8.

Action: We corrected fig. captions and added figures to the high std.dev. in Fig.8

  1. In FTIR spectra, the author should include the units for both the x and y axis

Action: We added units.

  1. It would be helpful if the authors could include a reference table along with exact peak values with respect to the FTIR spectra in Fig. 8.

Action: We added a reference value table (Table 3) and peak values.

  1. It would be helpful if a zoomed-in portion of the fingerprint region can be represented.

Action: We added a zoomed-in portion of the fingerprint region as Figure 11.

  1. The FTIR spectra should be baseline subtracted and deconvoluted mandatorily for better representation and understanding. There are several peaks that are not explained in the text or discussion.

Action: We explained the peaks in more detail in the discussion and by the newly added Table 3.

  1. Please mention the magnifications used for optical images.

Action: We added magnification to Figures 4 and 5.

  1. Is there any reference for 2.3 Biodegradation tests, and 2.4. Determination of moisture content and carboxyl group content? If so, please add it.

Action: We added references for moisture content and carboxyl group content.

  1. In general, how many times each experiment was repeated? author should mention.

Action: We have added number of repetitions for carboxylic group number and degree of polymerisation.

Round 2

Reviewer 1 Report

The authors made a number of corrections to the manuscript and took into account some of the reviewer's comments. I would like to draw the attention of the authors to the fact that the added references to the works of Th. Roder are not exhaustive, I recommend paying attention to this work - https://repositum.tuwien.at/handle/20.500.12708/50752 Lines 101-104. Please add a link. It is not entirely clear to me why it is necessary to use viscose fibers, which, as already shown in numerous works, are associated with a large environmentally harmful footprint. On the other hand, Lyocell fibers are obtained in a more acceptable way. It is also necessary to pay attention here that in order to obtain Lyocell fibers for the task described in the work, it is better to use not cellulose obtained from wood, but from waste, for example, flax waste (tow, etc.), which are formed in large quantities in a number of countries (France, Belgium, Russia, China, etc.) (example - https://doi.org/10.3390/fib10050045). At the very least, this issue should be noted in the manuscript ... Figure 2. Need to increase the font. Figure 7. What happens to sample P4, why does DP increase with time? Figure 10. For a sample of 35 weeks, a band is observed in the region of 1750, which is not observed on other samples, how do the authors explain this? references. It is desirable to arrange in accordance with the requirements of the journal with reference numbers.

Author Response

Response to reviewer – 2nd round -Polymers-1910437

Cellulose fibre degradation in cellulose/steel hybrid geotextiles under outdoor weathering conditions

Avinash P. Manian, Barbara Paul, Helene Lanter, Thomas Bechtold and Tung Pham

Dear Editor,

First we want to thank the reviewers for the careful revision. We considered all recommended changes and hope the manuscript now is acceptable for publication. We have highlighted the revised elements in colour.

Please find below the list of changes.

Thomas Bechtold

Reviewer 1

  1. The authors made a number of corrections to the manuscript and took into account some of the reviewer's comments. I would like to draw the attention of the authors to the fact that the added references to the works of Th. Roder are not exhaustive, I recommend paying attention to this work - https://repositum.tuwien.at/handle/20.500.12708/50752 Lines 101-104. Please add a link.

Action: We have added a comment and reference (page 2 last paragraph)

  1. It is not entirely clear to me why it is necessary to use viscose fibers, which, as already shown in numerous works, are associated with a large environmentally harmful footprint.

Action: We added a sentence in the introduction which explains that we have prepared the prototypes in cooperation with a viscose fibre producer. (page 3,  paragraph 4)

  1. On the other hand, Lyocell fibers are obtained in a more acceptable way. It is also necessary to pay attention here that in order to obtain Lyocell fibers for the task described in the work, it is better to use not cellulose obtained from wood, but from waste, for example, flax waste (tow, etc.), which are formed in large quantities in a number of countries (France, Belgium, Russia, China, etc.) (example - https://doi.org/10.3390/fib10050045). At the very least, this issue should be noted in the manuscript ...

Action: We added discussion and references in the conclusion section.

  1. Figure 2. Need to increase the font.

Action: We increased font in Figure 2

  1. Figure 7. What happens to sample P4, why does DP increase with time?

Action: We added a comment at page 11 last paragraph to discuss the variation in the results, which then also includes the value of P4.

  1. Figure 10. For a sample of 35 weeks, a band is observed in the region of 1750, which is not observed on other samples, how do the authors explain this?

Action: We added an explanation (plant material impurities and microbial growth) and a reference at page 19 first paragraph.

  1. It is desirable to arrange in accordance with the requirements of the journal with reference numbers.

Action: We arranged to reference numbers, we organised references style by Mendeley according to the “Journal of American Society” and hope this is acceptable

Reviewer 2 Report

The authors addressed all the comments. It can be accepted in present form.

Author Response

There were no further comments to improve the manuscript according to reviewer 2.